# Learning from Failure:
# Training Debiased Classifier from Biased Classifier

**Junhyun Nam**[1]     **Hyuntak Cha**[2]     **Sungsoo Ahn**[1]     **Jaeho Lee**[1]     **Jinwoo Shin**[1,2]

[1]School of Electrical Engineering, KAIST
[2]Graduate School of AI, KAIST
{junhyun.nam, hyuntak.cha, sungsoo.ahn, jaeho-lee, jinwoos}@kaist.ac.kr

## Abstract

Neural networks often learn to make predictions that overly rely on spurious corre-lation existing in the dataset, which causes the model to be *biased*. While previous work tackles this issue by using explicit labeling on the spuriously correlated attributes or presuming a particular bias type, we instead utilize a cheaper, yet generic form of human knowledge, which can be widely applicable to various types of bias. We first observe that neural networks learn to rely on the spurious correlation only when it is "easier" to learn than the desired knowledge, and such reliance is most prominent during the early phase of training. Based on the obser-vations, we propose a failure-based debiasing scheme by training a pair of neural networks simultaneously. Our main idea is twofold; (a) we intentionally train the first network to be biased by repeatedly amplifying its "prejudice", and (b) we debias the training of the second network by focusing on samples that go against the prejudice of the biased network in (a). Extensive experiments demonstrate that our method significantly improves the training of network against various types of biases in both synthetic and real-world datasets. Surprisingly, our framework even occasionally outperforms the debiasing methods requiring explicit supervision of the spuriously correlated attributes.

## 1   Introduction

When trained on carefully curated datasets, deep neural networks achieve state-of-the-art perfor-mances on many tasks in artificial intelligence, including image classification [11], object detection [8], and speech recognition [9]. On the other hand, neural networks often dramatically fail when trained on a highly biased dataset, by learning the unintended decision rule that works well only on the dataset being trained on. For instance, it is widely known that object classification datasets suffer from such misleading correlations [23, 29]. As an example, suppose that the "boat" is the only object category appearing in the images with the "water" background. When trained on a dataset with such bias, neural networks often learn to make predictions using the unintended decision rule based on the background of images, whereas a learner intended to learn the decision rule based on the object in the images.

To train a *debiased model* that captures the "intended correlation" from such biased datasets, recent approaches focus on how to utilize various types of human supervision effectively. One of the most popular forms of such supervision is an explicit label that indicates the misleadingly correlated attribute [14, 18, 24]. For instance, Kim et al. [14], Li and Vasconcelos [18] consider training a model to classify digits (instead of misleadingly correlated color) from the Colored MNIST dataset, under the setup where RGB values for coloring digits are given as side information. Another line of research focuses on developing algorithms tailored to a domain-specific type of bias in the target dataset, whose existence and characteristics are diagnosed by human experts [7, 25, 4, 2]. For instance,

Geirhos et al. [7] diagnose that ImageNet-trained classifiers are biased toward texture instead of a presumably more human-aligned notion of shapes, and use this takeaway to construct an augmented dataset to train shape-oriented classifiers.

Acquiring human supervision on the bias, however, is often a dauntingly laborious and expensive task. Gathering explicit labels of such misleadingly correlated attributes requires manual labeling by the workers that have a clear understanding of the underlying bias. Collecting expert knowledge of a human-perceived bias (e.g., texture bias) takes even more effort, as it requires a careful ablation study on the classifiers trained on biased datasets, e.g., Geirhos et al. [7] synthesize data via style transfer [6] to discover the existence of texture bias. Hence, an approach to train a debiased classifier without relying on such expensive supervision is warranted.

**Contribution.** In this paper, we propose a *failure-based* debiasing scheme, coined Learning from Failure (LfF). Our scheme does not require expensive supervision on the bias, such as explicit labels of misleadingly correlated attributes, or bias-tailored training technique. Instead, our method utilizes a cheaper form of human knowledge, leveraging the following intriguing observations on neural networks that are being trained on biased datasets.

We first observe that a biased dataset does not necessarily lead the model to learn the unintended decision rule; the bias negatively affects the model only when the bias attribute is "easier" to learn than the target attribute (Section 2.2). For the bias that negatively affects the model, we also observe that samples aligned with the bias show distinct loss trajectories in the training phase compared to samples conflicting with the bias. To be more specific, the classifier learns to fit samples aligned with the bias during the early stage of training and learns samples conflicting with the bias later (Section 2.3). The latter observation lines up with recent findings on training dynamics of deep neural networks [1, 21, 17]; networks tend to defer learning hard concepts, e.g., samples with random labels, to the later phase of training.

Based on the findings, we propose the following debiasing scheme, LfF. We simultaneously train two neural networks, one to be biased and the other to be debiased. Specifically, we train a "biased" neural network by amplifying its early-stage predictions. Here, we employ generalized cross entropy loss [28] for the biased model to focus on easy samples, which are expected to be samples aligned with bias. In parallel, we train a "debiased" neural network by focusing on samples that the biased model struggles to learn, which are expected to be samples conflicting with the bias. To this end, we re-weight training samples using the relative difficulty score based on the loss of the biased model and the debiased model (Section 3).

We show the effectiveness of LfF on various biased datasets, including Colored MNIST [14, 18] with color bias, Corrupted CIFAR-10 [12] with texture bias, and CelebA [19] with gender bias. In addition, we newly construct a real-world dataset, coined biased action recognition (BAR), to resolve the lack of realistic evaluation benchmark for debiasing schemes. In all of the experiments, our method succeeds in training a debiased classifier. In particular, our method improves the accuracy of the unbiased evaluation set by $35.34\% \rightarrow 63.39\%$, $17.93\% \rightarrow 31.66\%$, for the Colored MNIST and Corrupted CIFAR-10[1] datasets, respectively, even when $99.5\%$ of the training samples are bias-aligned (Section 4).

## 2 A closer look at training deep neural networks on biased datasets

In this section, we describe two empirical observations on training the neural networks with a biased dataset. These observations serve as a key intuition for designing and understanding our debiasing algorithm. We first provide a formal description of biased datasets in Section 2.1. Then we provide our empirical observations in Section 2.2 and Section 2.3.

### 2.1 Setup

Consider a dataset $\mathcal{D}$ where each input $x$ can be represented by a set of (possibly latent) *attributes* $\{a_1, \ldots, a_k\}$ for $a_i \in \mathcal{A}_i$ that describes the input. The goal is to train a predictor $f$ that belongs to a set of *intended decision rules* $\mathcal{F}_t$, consisting of decision rules that correctly predict the *target attribute* $y = a_t \in \mathcal{A}_t$. We say that a dataset $\mathcal{D}$ is *biased*, if (a) there exists another attribute $a_b \neq y$ that is highly correlated to the target attribute $y$ (i.e., $H(y|a_b) \approx 0$), and (b) one can settle an *unintended decision rule* $g_b \notin \mathcal{F}_t$ that correctly classifies $a_b$. We denote such an attribute $a_b$ by a *bias attribute*.

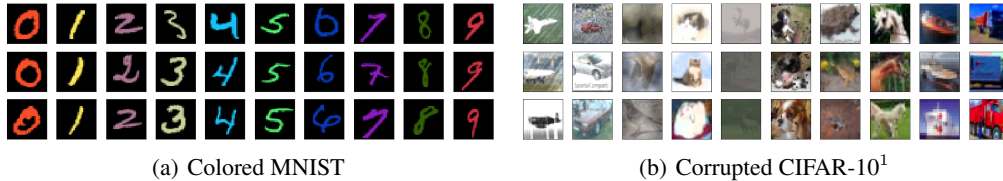

(a) Colored MNIST            (b) Corrupted CIFAR-10[1]

Figure 1: Illustration of bias-aligned samples for Colored MNIST, and Corrupted CIFAR-10[1] datasets.

Table 1: Accuracy of the unbiased evaluation set with varying choice of (target, bias) attribute pair for the Colored MNIST and Corrupted CIFAR-10[1,2] datasets. Accuracy and Accuracy* denotes the performance of the model trained on the biased and unbiased training set, respectively.

| Dataset | Target | Bias | Accuracy | Accuracy* | Relative drop |
|---|---|---|---|---|---|
| Colored MNIST | Color | Digit | $99.97_{\pm0.04}$ | $100.0_{\pm0.00}$ | -0.03% |
| | Digit | Color | $50.34_{\pm0.16}$ | $96.41_{\pm0.07}$ | -47.79% |
| Corrupted CIFAR-10[1] | Corruption | Object | $98.34_{\pm0.26}$ | $99.62_{\pm0.03}$ | -1.28% |
| | Object | Corruption | $22.72_{\pm0.87}$ | $80.00_{\pm0.01}$ | -71.60% |
| Corrupted CIFAR-10[2] | Corruption | Object | $98.64_{\pm0.20}$ | $99.80_{\pm0.01}$ | -1.16% |
| | Object | Corruption | $21.07_{\pm0.29}$ | $79.65_{\pm0.11}$ | -73.56% |

In biased datasets with a bias attribute $a_b$, we say that a sample is *bias-aligned* whenever it can be correctly classified by the unintended decision rule $g_b$, and *bias-conflicting* whenever it cannot be correctly classified by $g_b$.

Throughout the paper, we consider two types of evaluation datasets: the unbiased and bias-conflicting evaluation sets. We construct the unbiased evaluation set in a way that the target and bias attributes are uncorrelated. To this end, the unbiased evaluation set is constructed to have the same number of samples for every possible value of $(a_t, a_b)$. We simply construct the bias-conflicting evaluation set by excluding bias-aligned samples from the unbiased evaluation set.

Here, we illustrate the examples of biased datasets using the datasets considered for the experiment in Section 2.2 and 2.3, i.e., Colored MNIST and Corrupted CIFAR-10.

**Colored MNIST.** We inject color with random perturbation into the MNIST dataset [16] designed for digit classification, resulting in a dataset with two attributes: Digit and Color. In the case of $(a_t, a_b) = $ (Digit, Color), a set of intended decision rules $\mathcal{F}_t$ consists of decision rules that correctly classify images based on the Digit of the images. Here, a decision rule based on other attributes, e.g. Color, is considered as an unintended decision rule. Figure 1(a) illustrates examples of bias-aligned samples, which can be correctly classified by an unintended decision rule based on Color.

**Corrupted CIFAR-10.** This dataset is generated by corrupting the CIFAR-10 dataset [15] designed for object classification, following the protocols proposed by Hendrycks and Dietterich [12]. The resulting dataset consists of two attributes, i.e., category of the Object and type of Corruption used. Similar to the Colored MNIST dataset, this results in two possible choices for the target and bias attribute. We use two sets of protocols for corruption to build two datasets, namely the Corrupted CIFAR-10[1] and the Corrupted CIFAR-10[2] datasets. See Figure 1(b) for corruption-biased examples. A detailed description of the datasets is provided in Appendix A.

## 2.2 Not all biases are malignant

Our first observation is that training a classifier with a biased dataset does not necessarily lead to learning an unintended decision rule. Instead, the bias in the dataset negatively affects the prediction only if the bias is easier to be captured by the learned classifier. In Table 1, we report the accuracy of the classifiers for the unbiased evaluation set. We first note the existence of *benign* bias, i.e., there are cases where the classifier has not been affected by the bias. Particularly, when there is a degradation of accuracy with a certain choice of the target and bias attribute, the degradation does not occur with the choice made in reversed order.

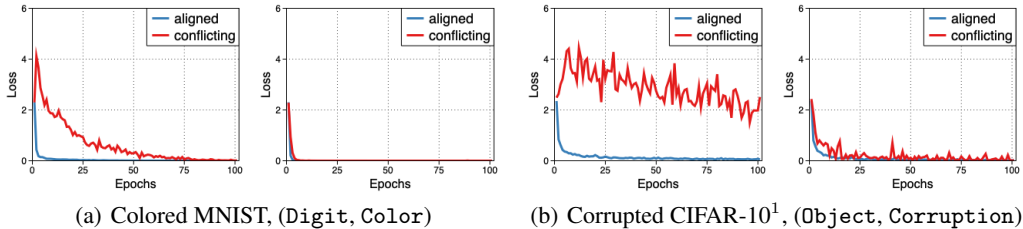

(a) Colored MNIST, (Digit, Color)     (b) Corrupted CIFAR-10[1], (Object, Corruption)

Figure 2: Illustration of neural network training on the biased datasets. For each dataset, the left and the right plots corresponds to training on the malignant bias and the benign bias. Our choice of (target, bias) attribute for the malignant bias is provided in brackets. For the benign bias, we choose the target and the bias attribute in reversed order.

From such an observation, we now define two types of bias: *malignant* and *benign*. For a biased dataset $\mathcal{D}$ with a target attribute $y$ and a bias attribute $a_b$, we say this bias is *malignant* if a model trained on $\mathcal{D}$ suffers performance degradation on unbiased evaluation set compared to one trained on another dataset which is not biased. In contrast, we say bias is *benign* if a model trained on $\mathcal{D}$ does not suffer such performance degradation. We interpret this observation as follows: the bias attribute inducing malignant bias is "easier" to learn than the target attribute, e.g., Color is easier to learn than Digit. To be specific, the classifier establishes its decision rule by relying on either (a) the intended correlation from the target or (b) the spurious correlation from the bias attribute. If (a) is harder to leverage than (b), the bias becomes malignant since the classifier learns unintended correlation. Otherwise, the classifier learns the correct correlation, hence the bias becomes benign.

### 2.3 Malignant bias is learned first

Next, we observe that the loss dynamics of bias-aligned samples and bias-conflicting samples during the training phase are in stark contrast, given that the bias is malignant. The loss of bias-aligned samples quickly declines to zero, but the loss of bias-conflicting samples first increases and starts decreasing after the loss of bias-aligned samples reaches near zero.

In Figure 2, we observe a significant difference between training dynamics on the malignant and the benign bias, consistently over the considered datasets. In the case of training under malignant bias, the training loss of the bias-conflicting samples is higher than the loss of the bias-aligned samples, and the gap is more significant during the early stages. In contrast, they are almost indistinguishable when trained under a benign bias.

We make an interesting connection between our observation and the observation made by Arpit et al. [1] on training neural networks on datasets with noisy labels. Arpit et al. [1] analyzed the training dynamics of the neural network on noisy datasets. They empirically demonstrate the preference of neural networks for easy concepts, e.g., common patterns in samples with correct labels, over the hard concepts, e.g., random patterns in samples with incorrect labels. One can interpret our results similarly; since the malignant bias attributes are easier to learn than the original task, the neural network tends to memorize it first.

## 3 Debiasing by learning from failure (LfF)

Based on our findings in Section 2, we propose a debiasing algorithm, coined Learning from Failure (LfF), for training neural networks on a biased dataset. At a high level, our algorithm simultaneously trains a pair of neural networks $(f_B, f_D)$ as follows: (a) intentionally training a model $f_B$ to be biased and (b) training a debiased model $f_D$ by focusing on the training samples that the biased model struggles to learn. In the rest of this section, we provide details on each component of our debiasing algorithm. We offer a full description of our debiasing scheme in Algorithm 1.

**Training a biased model.** We first describe how we train the biased model. i.e., the model following the unintended decision rule. From our observation in Section 2, we intentionally strengthen the prediction of the model from the early stage of training to make it follow the unintended decision rule. To this end, we use the following generalized cross entropy (GCE) [28] loss to amplify the bias

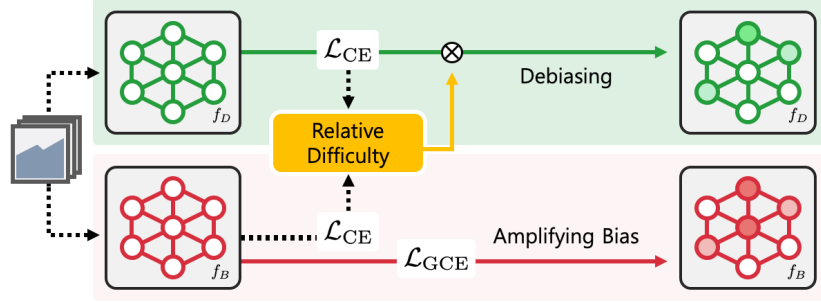

Figure 3: Illustration of training two models ($f_D$, $f_B$) to be debiased and biased, respectively. The biased model optimizes generalized cross entropy ($\mathcal{L}_{\text{GCE}}$) loss to amplify bias. The debiased model trains with weighted cross entropy loss leveraging relative difficulty. It results in larger weights to bias-conflicting samples while training the debiased model.

---

**Algorithm 1** Learning from Failure

1: **Input**: $\theta_B, \theta_D$, training set $\mathcal{D}$, learning rate $\eta$, number of iterations $T$
2: Initialize two networks $f_B(x; \theta_B)$ and $f_D(x; \theta_D)$.
3: **for** $t = 1, \cdots, T$ **do**
4:     Draw a mini-batch $\mathcal{B} = \{(x^{(b)}, y^{(b)})\}_{b=1}^B$ from $\mathcal{D}$
5:     Update $f_B(x; \theta_B)$ by $\theta_B \leftarrow \theta_B - \eta \nabla_{\theta_B} \sum_{(x,y) \in \mathcal{B}} \text{GCE}(f_B(x), y)$.
6:     Update $f_D(x; \theta_D)$ by $\theta_D \leftarrow \theta_D - \eta \nabla_{\theta_D} \sum_{(x,y) \in \mathcal{B}} \mathcal{W}(x) \cdot \text{CE}(f_D(x), y)$.
7: **end for**

---

of the neural network:

$$\text{GCE}(p(x; \theta), y) = \frac{1 - p_y(x; \theta)^q}{q}$$

where $p(x; \theta)$ and $p_y(x; \theta)$ are softmax output of the neural network and its probability assigned to the target attribute of $y$, respectively. Here, $q \in (0, 1]$ is a hyperparameter that controls the degree of amplification. For example, when $\lim_{q \to 0} \frac{1 - p^q}{q} = -\log p$, GCE becomes equivalent to standard cross entropy (CE) loss. Compared to the CE loss, the gradient of the GCE loss up-weights the gradient of the CE loss for the samples with a high probability $p_y$ of predicting the correct target attribute as follows:

$$\frac{\partial \text{GCE}(p, y)}{\partial \theta} = p_y^q \frac{\partial \text{CE}(p, y)}{\partial \theta},$$

Therefore, GCE loss trains a model to be biased by emphasizing the "easier" samples with the strong agreement between softmax output of the neural network and the target, which amplifies the "prejudice" of the neural network compare to the network trained with CE.

**Training a debiased model.** While we train a biased model as described earlier, we also train a debiased model simultaneously with the samples using the CE loss re-weighted by the following *relative difficulty* score:

$$\mathcal{W}(x) = \frac{\text{CE}(f_B(x), y)}{\text{CE}(f_B(x), y) + \text{CE}(f_D(x), y)},$$

where $f_B(x), f_D(x)$ are softmax outputs of the biased and debiased model, respectively. We use this score to indicate how much each sample is likely to be bias-conflicting to reflect our observation made in Section 2: for bias-aligned samples, biased model $f_B$ tends to have smaller loss compare to debiased model $f_D$ at the early stage of training, therefore having small weight for training debiased model. In other words, for bias-conflicting samples, biased model $f_B$ tends to have larger loss compare to debiased model $f_D$, result in large weight (close to 1) for training debiased model.

Table 2: Accuracy evaluated on unbiased samples for the Colored MNIST and Corrupted CIFAR-10[1,2] datasets with varying ratio of bias-aligned samples. We denote bias supervision type by ○ (no supervision), ◐ (bias-tailored supervision), and ● (explicit bias supervision). Best performing results are marked in bold.

| Dataset | Ratio (%) | Vanilla ○ | Ours ○ | HEX ◐ | REPAIR ● | Group DRO ● |
|---|---|---|---|---|---|---|
| Colored MNIST | 95.0 | $77.63_{\pm0.44}$ | $\mathbf{85.39}_{\pm0.94}$ | $70.44_{\pm1.41}$ | $82.51_{\pm0.59}$ | $84.50_{\pm0.46}$ |
| | 98.0 | $62.29_{\pm1.47}$ | $\mathbf{80.48}_{\pm0.45}$ | $62.03_{\pm0.24}$ | $72.86_{\pm1.47}$ | $76.30_{\pm1.53}$ |
| | 99.0 | $50.34_{\pm0.16}$ | $\mathbf{74.01}_{\pm2.21}$ | $51.99_{\pm1.09}$ | $67.28_{\pm1.69}$ | $71.33_{\pm1.76}$ |
| | 99.5 | $35.34_{\pm0.13}$ | $\mathbf{63.39}_{\pm1.97}$ | $41.38_{\pm1.31}$ | $56.40_{\pm3.74}$ | $59.67_{\pm2.73}$ |
| Corrupted CIFAR-10[1] | 95.0 | $45.24_{\pm0.22}$ | $\mathbf{59.95}_{\pm0.16}$ | $21.74_{\pm0.27}$ | $48.74_{\pm0.71}$ | $53.15_{\pm0.53}$ |
| | 98.0 | $30.21_{\pm0.82}$ | $\mathbf{49.43}_{\pm0.78}$ | $17.81_{\pm0.29}$ | $37.89_{\pm0.22}$ | $40.19_{\pm0.23}$ |
| | 99.0 | $22.72_{\pm0.87}$ | $\mathbf{41.37}_{\pm2.34}$ | $16.62_{\pm0.80}$ | $32.42_{\pm0.35}$ | $32.11_{\pm0.83}$ |
| | 99.5 | $17.93_{\pm0.66}$ | $\mathbf{31.66}_{\pm1.18}$ | $15.39_{\pm0.13}$ | $26.26_{\pm1.06}$ | $29.26_{\pm0.11}$ |
| Corrupted CIFAR-10[2] | 95.0 | $41.27_{\pm0.98}$ | $\mathbf{58.57}_{\pm1.18}$ | $19.25_{\pm0.81}$ | $54.05_{\pm1.01}$ | $57.92_{\pm0.31}$ |
| | 98.0 | $28.29_{\pm0.62}$ | $\mathbf{48.75}_{\pm1.68}$ | $15.55_{\pm0.84}$ | $44.22_{\pm0.84}$ | $46.12_{\pm1.11}$ |
| | 99.0 | $20.71_{\pm0.29}$ | $\mathbf{41.29}_{\pm2.08}$ | $14.42_{\pm0.51}$ | $38.40_{\pm0.26}$ | $39.57_{\pm1.04}$ |
| | 99.5 | $17.37_{\pm0.31}$ | $34.11_{\pm2.39}$ | $13.63_{\pm0.42}$ | $31.03_{\pm0.42}$ | $\mathbf{34.25}_{\pm0.74}$ |

Table 3: Accuracy evaluated on bias-conflicting samples for the Colored MNIST and Corrupted CIFAR-10[1,2] datasets with varying ratio of bias-aligned samples. We denote bias supervision type by ○ (no supervision), ◐ (bias-tailored supervision), and ● (explicit bias supervision). Best performing results are marked in bold.

| Dataset | Ratio (%) | Vanilla ○ | Ours ○ | HEX ◐ | REPAIR ● | Group DRO ● |
|---|---|---|---|---|---|---|
| Colored MNIST | 95.0 | $75.17_{\pm0.51}$ | $\mathbf{85.77}_{\pm0.66}$ | $67.75_{\pm1.49}$ | $83.26_{\pm0.42}$ | $83.11_{\pm0.41}$ |
| | 98.0 | $58.13_{\pm1.63}$ | $\mathbf{80.67}_{\pm0.56}$ | $58.80_{\pm0.28}$ | $73.42_{\pm1.42}$ | $74.28_{\pm1.93}$ |
| | 99.0 | $44.83_{\pm0.18}$ | $\mathbf{74.19}_{\pm1.94}$ | $46.96_{\pm1.20}$ | $68.26_{\pm1.52}$ | $69.58_{\pm1.66}$ |
| | 99.5 | $28.15_{\pm1.44}$ | $\mathbf{63.49}_{\pm1.94}$ | $35.05_{\pm1.46}$ | $57.27_{\pm3.92}$ | $57.07_{\pm3.60}$ |
| Corrupted CIFAR-10[1] | 95.0 | $39.42_{\pm0.20}$ | $\mathbf{59.62}_{\pm0.03}$ | $14.09_{\pm0.31}$ | $49.99_{\pm0.92}$ | $49.00_{\pm0.45}$ |
| | 98.0 | $22.65_{\pm0.95}$ | $\mathbf{48.69}_{\pm0.70}$ | $9.34_{\pm0.41}$ | $38.94_{\pm0.20}$ | $35.10_{\pm0.49}$ |
| | 99.0 | $14.24_{\pm1.03}$ | $\mathbf{39.55}_{\pm2.56}$ | $8.37_{\pm0.56}$ | $33.05_{\pm0.36}$ | $28.04_{\pm1.18}$ |
| | 99.5 | $10.50_{\pm0.71}$ | $\mathbf{28.61}_{\pm1.25}$ | $6.38_{\pm0.08}$ | $26.52_{\pm0.94}$ | $24.40_{\pm0.28}$ |
| Corrupted CIFAR-10[2] | 95.0 | $34.97_{\pm1.06}$ | $\mathbf{58.64}_{\pm1.04}$ | $10.79_{\pm0.90}$ | $54.46_{\pm1.02}$ | $54.60_{\pm0.11}$ |
| | 98.0 | $20.52_{\pm0.73}$ | $\mathbf{48.99}_{\pm1.61}$ | $6.60_{\pm7.23}$ | $44.63_{\pm0.75}$ | $42.71_{\pm1.24}$ |
| | 99.0 | $12.11_{\pm0.29}$ | $\mathbf{40.84}_{\pm2.06}$ | $5.11_{\pm0.59}$ | $38.81_{\pm0.20}$ | $37.07_{\pm1.02}$ |
| | 99.5 | $10.01_{\pm0.01}$ | $\mathbf{32.03}_{\pm2.51}$ | $4.22_{\pm0.43}$ | $31.45_{\pm0.28}$ | $30.92_{\pm0.86}$ |

## 4 Experiments

In this section, we demonstrate the effectiveness of our LfF algorithm proposed in Section 3, and introduce our newly constructed biased action recognition dataset, coined BAR. All experimental results in this section support that LfF successfully trains a debiased classifier, with the knowledge that the bias attribute is learned earlier than the target attribute (instead of explicit supervision for the bias present in the dataset). The classifiers trained by LfF consistently outperforms the vanilla classifiers (trained without any debiasing procedure) on both the unbiased and bias-conflicting evaluation set.

For the experiments in this section, we use MLP with three hidden layers, ResNet-20, and ResNet-18 [11] for the Colored MNIST, Corrupted CIFAR-10, and {CelebA, BAR} datasets, respectively. All results reported in this section are averaged over three independent trials. We provide a detailed description of datasets we considered in Appendix A, B and experimental details in Appendix C.

### 4.1 Controlled experiments

**Baselines.** For controlled experiments, we compare the performance of our algorithm with other debiasing algorithms, either presuming a particular bias type or requiring access to additional labels

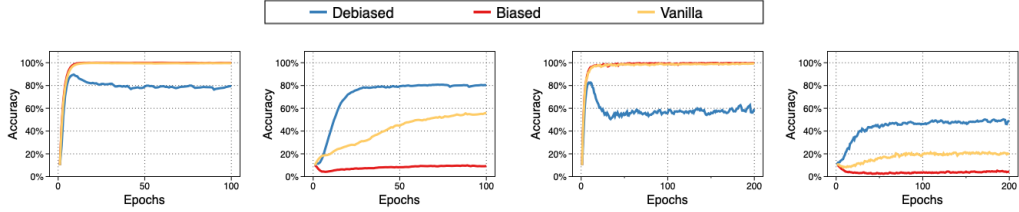

(a) Bias-{aligned, conflicting} Colored MNIST      (b) Bias-{aligned, conflicting} Corrupted CIFAR-10[1]

Figure 4: Learning curves of vanilla, biased, and debiased model for Colored MNIST and Corrupted CIFAR-10[1]. For each dataset, the left and the right plots correspond to curves for the bias-aligned samples and the bias-conflicting samples, respectively.

Table 4: Accuracy evaluated on the unbiased samples and bias-conflicting samples of the Colored MNIST and Corrupted CIFAR-10[1,2] dataset for ablation of the proposed algorithm. We denote our algorithm using vanilla model as biased model by Ours[†]. Best performing results are marked in bold.

| Dataset | Unbiased | | | Bias-conflicting | | |
|---|---|---|---|---|---|---|
| | Vanilla | Ours[†] | Ours | Vanilla | Ours[†] | Ours |
| Colored MNIST | $50.34_{\pm0.16}$ | $49.90_{\pm1.67}$ | $\mathbf{74.01}_{\pm2.21}$ | $44.83_{\pm0.18}$ | $44.44_{\pm1.83}$ | $\mathbf{74.19}_{\pm1.94}$ |
| Corrupted CIFAR-10[1] | $22.72_{\pm0.87}$ | $25.15_{\pm0.63}$ | $\mathbf{41.37}_{\pm2.34}$ | $14.24_{\pm1.03}$ | $17.21_{\pm0.69}$ | $\mathbf{39.55}_{\pm2.56}$ |
| Corrupted CIFAR-10[2] | $20.71_{\pm0.29}$ | $22.90_{\pm0.47}$ | $\mathbf{41.29}_{\pm2.08}$ | $12.11_{\pm0.29}$ | $14.89_{\pm0.66}$ | $\mathbf{40.84}_{\pm2.06}$ |

containing information about bias attributes. We consider HEX proposed by Wang et al. [25], which attempts to remove texture bias using the bias-specific knowledge. We also consider REPAIR and Group DRO proposed by Li and Vasconcelos [18] and Sagawa et al. [24], requiring explicit labeling of the bias attributes. We provide a detailed description of the employed baselines in Appendix D.

**Ratio of the bias-aligned samples.** In the first set of experiments, we vary the ratio of bias-aligned samples in the training dataset by selecting from $\{95.0\%, 98.0\%, 99.0\%, 99.5\%\}$. We experiment on Colored MNIST, and Corrupted CIFAR-10[1,2] datasets with $(a_t, a_b) = (\texttt{Digit}, \texttt{Color})$, and $(a_t, a_b) = (\texttt{Object}, \texttt{Corruption})$, respectively.

In Table 2, 3, we report the accuracy evaluated on unbiased samples and bias-conflicting samples. We observe that the proposed method significantly outperforms the baseline on all levels of ratio for bias-aligned samples. Most notably, LfF achieves 41.37% accuracy on the unbiased evaluation set for the Corrupted CIFAR-10[1] dataset with 99% bias-aligned samples, while the vanilla model only achieves 22.72%. In addition, we report the accuracy of unbiased evaluation set and bias-conflicting samples with varying levels of difficulty for the bias attributes in Appendix E.

**Detailed analysis of failure-based debiasing.** We further analyze the specific details of our algorithm. To be precise, we first investigate the accuracies of the vanilla model and the biased, debiased models $f_B, f_D$ trained by our algorithm for the bias-aligned and bias-conflicting samples. In Figure 4, we plot the training curves of each model. We observe both the vanilla model and the biased model easily achieve 100% accuracy for the bias-aligned samples. On the other hand, the vanilla model shows 50% accuracy for the bias-conflicting samples, and the biased model performs close to random guessing. From this result, we can say that our intentionally biased model only exploits the bias attribute without learning the target attribute.

To compare our debiased model to the vanilla model, we start by observing the accuracy gap between the bias-aligned and bias-conflicting samples. As described before, while the vanilla model easily achieves 100% accuracy for the bias-aligned samples, it cannot achieve similar performance for the bias-conflicting samples. In contrast, our debiased model shows consistent performance (about 80%) for both the bias-aligned and bias-conflicting samples. As a result, it indicates that our debiased model successfully learns the *intended* target attribute, while the predictions of the vanilla model heavily rely on the *unintended* bias attribute.

**Contribution of GCE loss.** We also test a variant of our algorithm, where we train the biased model with standard cross entropy instead of GCE. As observed in Figure 4, the model trained with standard cross entropy, denoted by Vanilla, not only exploits the bias attribute but also partially learns the target attribute. Therefore, one can expect that using such a CE-trained model as the biased model can hurt debiasing ability of our algorithm. In Table 4, we report the test performance of our debiased

Table 5: Accuracy evaluated on the unbiased samples and bias-conflicting samples for the CelebA dataset with `Gender` as the bias attribute.

| Target attribute | Unbiased | | | Bias-conflicting | | |
|---|---|---|---|---|---|---|
| | Vanilla | Ours | Group DRO | Vanilla | Ours | Group DRO |
| `HairColor` | $70.25_{\pm0.35}$ | $84.24_{\pm0.37}$ | $85.43_{\pm0.53}$ | $52.52_{\pm0.19}$ | $81.24_{\pm1.38}$ | $83.40_{\pm0.67}$ |
| `HeavyMakeup` | $62.00_{\pm0.02}$ | $66.20_{\pm1.21}$ | $64.88_{\pm0.42}$ | $33.75_{\pm0.28}$ | $45.48_{\pm4.33}$ | $50.24_{\pm0.68}$ |

Table 6: Class-wise action recognition accuracy on BAR evaluation set. Best performing results are marked in bold.

| Action | Climbing | Diving | Fishing | Racing | Throwing | Vaulting | Average |
|---|---|---|---|---|---|---|---|
| Vanilla | $59.05_{\pm17.48}$ | $16.56_{\pm1.58}$ | $62.69_{\pm3.64}$ | $77.27_{\pm2.62}$ | $28.62_{\pm2.95}$ | $66.92_{\pm7.25}$ | $51.85_{\pm5.92}$ |
| ReBias | $77.78_{\pm8.32}$ | $\mathbf{51.57}_{\pm4.54}$ | $54.76_{\pm2.38}$ | $80.56_{\pm2.19}$ | $28.63_{\pm2.71}$ | $65.14_{\pm7.21}$ | $59.74_{\pm1.49}$ |
| Ours | $\mathbf{79.36}_{\pm4.79}$ | $34.59_{\pm2.26}$ | $\mathbf{75.39}_{\pm3.63}$ | $\mathbf{83.08}_{\pm1.90}$ | $\mathbf{33.72}_{\pm0.68}$ | $\mathbf{71.75}_{\pm3.32}$ | $\mathbf{62.98}_{\pm2.76}$ |

model trained with the CE-trained biased model instead of the GCE-trained model, denoted as Ours$^\dagger$. As expected, using the CE-trained model to compute relative difficulty does not help debiasing model.

## 4.2 Real-world experiments

**CelebA.** The CelebA dataset [19] is a multi-attribute dataset for face recognition, equipped with 40 types of attributes for each image. Among 40 attributes, we find that `Gender` and `HairColor` attributes have a high correlation. Moreover, we observe the attribute `Gender` is used as a cue for predicting the attribute `HairColor`. Therefore, we use `HairColor` as the target and `Gender` as the bias attribute. Similarly, we do the same thing for `HeavyMakeup` as the target and `Gender` as the bias attribute.

In Table 5, we observe our algorithm consistently outperforms the vanilla model while the vanilla model suffers from gender bias existing in the real-world dataset. The accuracy gap between the vanilla model and our model is larger for the bias-conflicting samples, which indicates the vanilla model fails to learn the intended target attribute, instead exploits the biased statistic of the dataset. Notably, our model is comparable to Group DRO, which requires explicit labeling for the bias attribute (unlike ours), and even outperforms on the unbiased evaluation set when the target attribute is `HeavyMakeup`. This is because Group DRO aims to maximize the worst-case group accuracy, not overall unbiased accuracy.

**Biased action recognition dataset.** To verify effectiveness of our proposed scheme in a realistic setting, we construct a place-biased action recognition (BAR) dataset with training and evaluation set. We settle six typical action-place pairs by inspecting imSitu dataset [27], which provides action and place labels. We assign images describing these six typical action-place pairs to the training set and, otherwise, the evaluation set of BAR. BAR is publicly available[1], and a detailed description of BAR is in Appendix B.

BAR aims to resolve the lack of realistic evaluation benchmark for debiasing schemes. Since previous debiasing schemes have tackled certain types of bias, they have assumed the correlation between the target and bias attribute to be tangible, which indeed is not available in the case of real-world settings. Such an assumption also makes it hard to verify whether one's scheme can be applied to a wide range of realistic settings. BAR instead offers evaluation set which is constructed based on the intuition that "a majority of samples that do not match typical action-place pairs are bias-conflicting." To demonstrate, the evaluation set consists of samples not matching the settled six typical pairs. In the end, we can use this evaluation set, which would be similar to a set of bias-conflicting samples to verify our algorithm's effectiveness.

Table 6 illustrates the test accuracy on the BAR evaluation set. LfF outperforms the vanilla classifier for all action classes. This indicates that LfF encourages the model to train on relatively hard training samples, thereby leading to debiasing the model. In addition, LfF outperforms ReBias [2], which is also free from explicit labeling on the bias attribute, for most action classes except `Diving`.

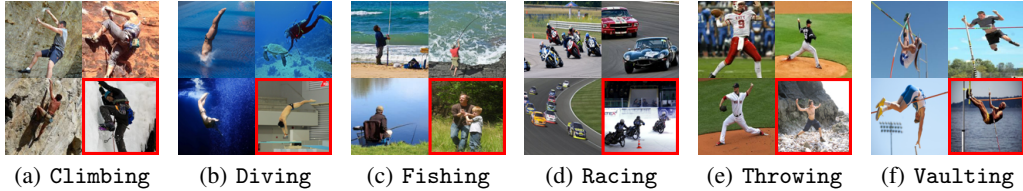

|(a) Climbing|(b) Diving|(c) Fishing|(d) Racing|(e) Throwing|(f) Vaulting|

Figure 5: Illustration of BAR images of six typical action-place pairs settled as (`Climbing`, `RockWall`), (`Diving`, `Underwater`), (`Fishing`, `WaterSurface`), (`Racing`, `APavedTrack`), (`Throwing`, `PlayingField`), and (`Vaulting`, `Sky`). The images with red border lines belong to BAR evaluation set, and others belong to BAR training set.

## 5 Related work

**Debiasing without explicit supervision.** In a real-world scenario, bias presented in the dataset often hard to be easily characterized in the form of labels. Even if one can characterize the bias, acquiring explicit supervision for the bias is still an expensive task that requires manual labeling by human labelers, having a clear understanding of the bias. To address this issue, there have been several works to resolve dataset bias, without explicit supervision on the bias. Geirhos et al. [7] observe the presence of texture bias in the ImageNet-trained classifiers, and train shape-oriented classifier with augmented data using style transfer [6]. Wang et al. [25] also aim to remove texture bias, using a hand-crafted module to extract bias and remove captured bias by domain adversarial loss and subspace projection. More recently, Bahng et al. [2] utilize the small-capacity model to capture bias and force debiased model to learn independent feature from the biased model.

**Debiasing from the biased model.** To train a debiased model, recent works utilize an intentionally biased model to debias another model. These works mainly focused on removing well-known dataset bias that can easily be characterized. Cadene et al. [3] use a question-only model to reduce question bias in a visual question answering (VQA) model. Clark et al. [5] construct bias-only models for the task having prior knowledge of existing biases, including VQA, reading comprehension, and natural language inference (NLI). Concurrently, He et al. [10] also train a biased model that only uses features known to relate to dataset bias in NLI. While these works are limited to biases existing in the NLP domain, Bahng et al. [2] capture local texture bias in image classification and static bias in video action recognition task using small-capacity models.

Previous works mentioned above have leveraged expert knowledge, used as a substitute for explicit supervision, for a particular type of human-perceived bias. We, in a more straightforward approach, consider general properties of bias from observations on training dynamics of bias-aligned and bias-conflicting samples. Although our method also uses a certain form of human knowledge that whether existing bias in the dataset follows our observation, this is a yes/no type of knowledge, which has advantages in its affordability and applicability. We also assume the existence of bias-conflicting samples on which the debiased model should focus, which indeed is the case in real-life application scenarios. In the end, we propose a simple yet widely applicable debiasing scheme free from the choice for form and amount of supervision on the bias.

## 6 Conclusion

In this work, we propose a debiasing scheme, coined Learning from Failure (LfF), for training neural networks in the biased dataset. Our framework is based on an important observation on relationship between the training of neural networks and the "easiness" of biased attribute. Through extensive experiments, LfF shows successful results on the debiased training of neural networks. We expect our achievements may shed light on the nature of debiasing neural networks with minimal human supervision.

## Broader Impact

Mitigating the potential risk caused by biased datasets is a timely subject, especially with the widespread use of AI systems in our daily lives. Since the world is biased by nature, biased models are often deployed without perceiving their discriminative behavior, thereby leading to invoking the potential risk. For instance, a facial recognition software in digital cameras turned out to "over-predict" Asians as blinking when it was trained on Caucasian faces [20]. Disregarding such potential risks would further result in critical social issues [13], such as racism, gender discrimination, filter bubbles [22], and social polarization.

We propose the debiasing scheme to mitigate the aforementioned potential risks. A common approach to reducing the risks is to develop schemes that specifically tackle a bias of interest, e.g., gender, race, etc. However, underexplored biases might exist in the dataset, but bias-specific schemes would not be able to address these other biases. We thus recommend leveraging the general behaviors of neural networks trained on biased datasets, which can be applied for debiasing in diverse applications.

Now we discuss potential benefits and limitations of the proposed scheme. The underexplored types of bias can be discovered by using a set of samples that are hard for the model to learn. Using this approach can increase awareness of underexplored biases. This awareness can be specifically important for groups that would be potentially affected. We acknowledge that assessing the reduction of potential risks by the proposed scheme can be a challenge without specifically identifying the biases. Still, we anticipate that our approach opens a potential to analyze and interpret underexplored types of bias.

## Acknowledgements

This work was supported by Institute of Information & communications Technology Planning & Evaluation (IITP) grant funded by the Korea government(MSIT) (No.2019-0-00075, Artificial Intelligence Graduate School Program (KAIST) and No.2019-0-01396, Development of framework for analyzing, detecting, mitigating of bias in AI model and training data)

## Footnotes

[1] https://github.com/alinlab/BAR

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
