[Supplementary Material]

# A Datasets

## A.1 Controlled experiments

We conduct the experiments on biased datasets with the same number of categories for the target and bias attributes, i.e., $|\mathcal{A}_t| = |\mathcal{A}_b|$. Furthermore, the empirical distribution $p_{\texttt{train}}$ of the training dataset is biased to satisfy the following equation:

$$p_{\texttt{train}}(a_b|a_t) = \begin{cases} p_{\texttt{align}} & \text{if } g(a_t) = a_b, \\ \dfrac{p_{\texttt{conflict}}}{|\mathcal{A}_b| - 1} & \text{otherwise}, \end{cases}$$

$$p_{\texttt{align}} > p_{\texttt{conflict}}, \quad \sum_{a_b \in \mathcal{A}_b} p_{\texttt{train}}(a_b|a_t) = 1.$$

Here, $a_t$ and $a_b$ are the target and bias attribute, respectively. The function $g : \mathcal{A}_t \to \mathcal{A}_b$ is a bijection between the target and bias attribute that assigns the bias attribute to each value of the target attribute. $p_{\texttt{align}}$, $p_{\texttt{conflict}}$ are the ratio of bias-aligned and bias-conflicting samples, respectively. In what follows, we describe instance-specific details on the datasets considered in the experiments.

**Colored MNIST.** The MNIST dataset [16] consists of grayscale digit images. We modify the original MNIST dataset to have two attributes: `Digit` and `Color`. Note that similar modification has been proposed by Kim et al. [14], Li and Vasconcelos [18], Bahng et al. [2]. To define the `Color` attribute, we first choose ten distinct RGB values by drawing them uniformly at random. We use these ten RGB values throughout all the experiments for the Colored MNIST dataset. Then we generate ten `Color` distributions by assigning chosen RGB values to each `Color` distribution as its mean. Each `Color` distribution is a 3-dimensional Gaussian distribution having the assigned RGB value as its mean with predefined covariance $\sigma^2 I$. We pair `Digit` $a_t$ and `Color` distribution $a_b$ to make a correlation between two attributes, `Digit` and `Color`. Each bias-aligned sample has a `Digit` colored by RGB value sampled from paired `Color` distribution, and each bias-conflicting sample has a `Digit` colored by RGB value sampled from the other (nine) `Color` distributions. We control the ratio of bias-aligned samples among $\{99.5\%, 99.0\%, 98.0\%, 95.0\%\}$. The level of difficulty for the bias attribute is defined by the variance ($\sigma^2$) of the `Color` distribution. We vary the standard deviation ($\sigma$) of the `Color` distributions among $\{0.05, 0.02, 0.01, 0.005\}$. We use 60,000 training samples and 10,000 test samples.

**Corrupted CIFAR-10.** This dataset is generated by corrupting the CIFAR-10 dataset [15] designed for object classification, following the protocols proposed by Hendrycks and Dietterich [12]. The resulting dataset consists of two attributes, i.e., category of the `Object` and type of `Corruption` used. We use two sets of protocols for `Corruption` to build two datasets, namely the Corrupted CIFAR-10[1] and the Corrupted CIFAR-10[2] datasets. In particular, the Corrupted CIFAR-10[1,2] datasets use the following types of `Corruption`, respectively: {`Snow`, `Frost`, `Fog`, `Brightness`, `Contrast`, `Spatter`, `Elastic`, `JPEG`, `Pixelate`, `Saturate`} and {`GaussianNoise`, `ShotNoise`, `ImpulseNoise`, `SpeckleNoise`, `GaussianBlur`, `DefocusBlur`, `GlassBlur`, `MotionBlur`, `ZoomBlur`, `Original`}, respectively. In order to introduce the varying levels of difficulty, we control the "severity" of `Corruption`, which was predefined by Hendrycks and Dietterich [12]. As the `Corruption` gets more severe, the images are likely to lose their characteristics and become less distinguishable. We use 50,000 training samples and 10,000 test samples for this dataset.

## A.2 Real-world experiments

**CelebA.** The CelebA dataset [19] is a multi-attribute dataset for face recognition, equipped with 40 types of attributes for each image. Among 40 attributes, we use the `BlondHair` attribute (denoted by `HairColor` in the main text) following Sagawa et al. [24], and additionally consider `HeavyMakeup` attribute as the target attributes. For both of the cases, we use `Male` attribute (denoted by `Gender` in the main text) as the bias attribute. The dataset consists of 202,599 face images, and we use the official train-val split for training and test (162,770 for training, 19,867 for test). To evaluate the unbiased accuracy with an imbalanced evaluation set, we evaluate accuracy for each value of $(a_t, a_b)$, and compute average accuracy over all $(a_t, a_b)$ pairs.

# B    Biased action recognition dataset

**6-class action recognition dataset.** Biased Action Recognition (BAR) dataset is a real-world image dataset categorized as six action classes which are biased to distinct places. We carefully settle these six action classes by inspecting imSitu [27], which provides still action images from Google Image Search with action and place labels. In detail, we choose action classes where images for each of these candidate actions share common place characteristics. At the same time, the place characteristics of action class candidates should be distinct in order to classify the action only from place attributes. In the end, we settle the six typical action-place pairs as (`Climbing`, `RockWall`), (`Diving`, `Underwater`), (`Fishing`, `WaterSurface`), (`Racing`, `APavedTrack`), (`Throwing`, `PlayingField`), and (`Vaulting`, `Sky`).

**The source of dataset.** We construct BAR with images from various sources: imSitu [27], Stanford 40 Actions [26], and Google Image Search. In the case of imSitu [27], we merge several action classes where the images have a similar gesture for constructing a single action class of BAR dataset, e.g., {`hurling`, `pitching`, `flinging`} for constructing `throwing`, and {`carting`, `skidding`} for constructing `racing`.

**Construction process.** BAR consists of training and evaluation sets; images describing the typical six action-place pairs belong to the training set and otherwise, the evaluation set. Before splitting images into these two sets, we exclude inappropriate images: illustrations, clip-arts, images with solid color background, and different gestures with the target gesture of the settled six action-place pairs. Since our sanitized images do not have explicit place labels, we split images into two sets by workers on Amazon Mechanical Turk. We designed the reasoning process to help workers answer the given questions. To be more specific, workers were asked to answer three binary questions. We split images into 'invalid', 'training', and 'evaluation' set based on workers' responses through binary questions. Workers were also asked to draw a bounding box where they considered it a clue to determine the place in order to help workers filter out images without an explicit clue. Here is the list of binary questions for each action class:

Figure 6: Illustration of BAR reasoning process.

- `Climbing`: *Does the picture clearly describe* `Climbing` *and include person?*, *Then, is the person rock climbing?*, *Draw a box around the natural rock wall (including a person) on the image. If it is not natural rock wall, click 'Cannot find clue'.*

- `Diving`: *Does the picture clearly describe* `Scuba Diving / Diving jump / Diving` *and include person?*, *Then, does the picture include a body of water or the surface of a body of water?*, *Draw a box around a body of water or the surface of a body of water (including a person) on the image.*

- `Fishing`: *Does the picture clearly describe* `Fishing` *and include person?*, *Then, does the picture contain the surface of a body of a water?*, *Draw a box around the surface of a body of a water (including a person) on the image. If the water region does not more than 90% of the image's background, click 'Cannot find clue'.*

- `Racing`: *Does the picture clearly describe* `Auto racing / Motorcycle racing / Cart racing ?*, *Then, is the racing held on a paved track?*, *Draw a box around a paved track (including a vehicle) on the image.*

- `Throwing`: *Does the picture clearly capture the* `Throwing / Pelting` *moment and include person?*, *Then, can you see a type of playing field (baseball mound, football pitch, etc.) where the person is throwing something on?*, *Draw a box around the playing field (baseball mound, football pitch, etc.) on the image.*

- `Vaulting`: *Does the picture clearly capture the* `Pole Vaulting` *and include person?*, *Then, does the picture contain the sky as background? Draw a box around the sky region (including a person) on the image. If the sky region does not more than 90% of the image's background, click 'Cannot find clue'.*

Finally, we use 2,595 images to construct BAR dataset. All image sizes are over 400px width and 300px height. The BAR training and evaluation sets are publicly available on `https://anonymous.4open.science/r/c9025a07-2784-47fb-8ba1-77b06c3509fe/`.

Table 7: Per-class count of BAR dataset.

| Action | Climbing | Diving | Fishing | Racing | Throwing | Vaulting | Total |
|---|---|---|---|---|---|---|---|
| Training | 326 | 520 | 163 | 336 | 317 | 279 | 1941 |
| Evaluation | 105 | 159 | 42 | 132 | 85 | 131 | 654 |

(a) *Describe action?*

(b) *Match typical pairs?*

(c) *Draw a bounding box.*

(d) *Finish*

Figure 7: Overview of web pages for workers to validate and split images of BAR. Workers are asked to answer three binary questions and drawing a bounding box task.

# C  Experimental details

**Architecture details.** For the Colored MNIST dataset, we use the multi-layered perceptron consisting of three hidden layers where each hidden layer consists of 100 hidden units. For the Corrupted CIFAR-10 dataset, we use the ResNet-20 proposed by He et al. [11]. For CelebA and BAR, we employ the Pytorch `torchvision` implementation of the ResNet-18 model, starting from pretrained weights.

**Training details.** We use Adam optimizer throughout all the experiments in the paper. We use a learning rate of $0.001$ and a batch size of $256$ for the Colored MNIST and Corrupted CIFAR-10 datasets. We use a learning rate of $0.0001$ and a batch size of $256$ for the CelebA and BAR dataset. Samples were augmented with random crop and horizontal flip transformations for the Corrupted CIFAR-10 and BAR dataset, and horizontal flip transformation for CelebA. For the Corrupted CIFAR-10 dataset, we take $32 \times 32$ random crops from image padded by 4 pixels on each side. For the BAR dataset, we take $224 \times 224$ random crops using `torchvision.transforms.RandomResizedCrop` in Pytorch. We do not use data augmentation schemes for training the neural network on the Colored MNIST dataset. We train the networks for 100, 200, 50, and 90 epochs for Colored MNIST, Corrupted CIFAR-10, CelebA and BAR, respectively. The GCE hyperparamter $q = 0.7$ is simply taken from the original paper [28]. For stable training of LfF, we use an exponential moving average of loss for computing relative difficulty score instead of loss at each training epoch, with a fixed exponential decay hyperparameter $0.7$.

# D  Baselines

(1) HEX [25] attempts to mitigate texture bias when the texture related domain identifier is not available. By utilizing *gray-level co-occurrence matrix* (GLCM), neural gray-level co-occurrence Matrix (NGLCM) can capture superficial statistics on the images, and HEX projects the model's representation orthogonal to the captured texture bias. Since our interest in debiasing is similar to that of HEX in terms of a method without explicit supervision on the bias, we use HEX as a baseline to compare debiasing performance in the case of controlled experiments.

(2) REPAIR [18] "re-weights" training samples to have minimal mutual information between the bias-relevant labels and the intermediate representations of the target classifier. We test all four variants of REPAIR: REPAIR-T, REPAIR-R, REPAIR-C, and REPAIR-S, corresponding to the re-weighting schemes based on thresholding, ranking, per-class ranking, and sampling, respectively. We report the best result among four variants of REPAIR. We use RGB values for coloring digits and classes of corruption as representations inducing bias for the Colored MNIST and Corrupted CIFAR-10 datasets, respectively.

(3) Group DRO [24] aims to minimize "worst-case" training loss over a set of pre-defined groups. Note that one requires additional labels of the bias attribute to define groups to apply group DRO for our problem of interest. With the label of bias attribute, we define $|\mathcal{A}_t| \times |\mathcal{A}_b|$ groups, one for each value of $(a_t, a_b)$. Sagawa et al. [24] expect that models that learn the spurious correlation between $a_t$ and $a_b$ in the training data would do poorly on groups for which the correlation does not hold, and hence do worse on the worst-group.

# E  Additional experiments

Table 8: Accuracy evaluated on the unbiased samples for the Colored MNIST and Corrupted CIFAR-10[1,2] datasets with varying difficulty of the bias attributes. We denote bias supervision type by ○ (no supervision), ◐ (bias-tailored supervision), and ● (explicit bias supervision). Best performing results are marked in bold.

| Dataset | Difficulty | Vanilla ○ | Ours ○ | HEX ◐ | REPAIR ● | Group DRO ● |
|---|---|---|---|---|---|---|
| Colored MNIST | 1 | $50.97_{\pm0.59}$ | $\mathbf{75.91}_{\pm1.25}$ | $51.38_{\pm0.59}$ | $69.60_{\pm0.97}$ | $70.34_{\pm1.98}$ |
| | 2 | $50.92_{\pm1.16}$ | $\mathbf{74.05}_{\pm2.21}$ | $51.38_{\pm0.59}$ | $64.14_{\pm0.38}$ | $70.80_{\pm1.82}$ |
| | 3 | $49.66_{\pm0.42}$ | $\mathbf{72.50}_{\pm1.79}$ | $52.88_{\pm1.24}$ | $69.20_{\pm2.03}$ | $71.03_{\pm2.24}$ |
| | 4 | $50.34_{\pm0.16}$ | $\mathbf{74.01}_{\pm2.21}$ | $51.99_{\pm1.09}$ | $67.28_{\pm1.69}$ | $71.33_{\pm1.76}$ |
| Corrupted CIFAR-10[1] | 1 | $35.37_{\pm0.58}$ | $\mathbf{52.12}_{\pm1.99}$ | $23.92_{\pm0.80}$ | $37.73_{\pm0.73}$ | $49.62_{\pm1.49}$ |
| | 2 | $29.30_{\pm3.11}$ | $\mathbf{47.19}_{\pm2.26}$ | $21.23_{\pm0.38}$ | $36.22_{\pm0.88}$ | $44.54_{\pm1.70}$ |
| | 3 | $26.44_{\pm0.98}$ | $\mathbf{44.12}_{\pm1.53}$ | $18.66_{\pm1.16}$ | $34.59_{\pm1.88}$ | $38.43_{\pm1.44}$ |
| | 4 | $22.72_{\pm0.87}$ | $\mathbf{41.37}_{\pm2.34}$ | $16.62_{\pm0.80}$ | $32.42_{\pm0.35}$ | $32.11_{\pm0.83}$ |
| Corrupted CIFAR-10[2] | 1 | $32.00_{\pm0.87}$ | $\mathbf{46.89}_{\pm3.02}$ | $20.12_{\pm0.44}$ | $41.00_{\pm0.39}$ | $44.85_{\pm0.04}$ |
| | 2 | $27.62_{\pm1.31}$ | $\mathbf{43.56}_{\pm2.10}$ | $16.82_{\pm0.38}$ | $39.57_{\pm0.61}$ | $43.21_{\pm1.54}$ |
| | 3 | $22.14_{\pm0.03}$ | $41.46_{\pm0.30}$ | $15.22_{\pm0.47}$ | $38.16_{\pm0.52}$ | $\mathbf{42.12}_{\pm0.52}$ |
| | 4 | $20.71_{\pm0.29}$ | $\mathbf{41.29}_{\pm2.08}$ | $14.42_{\pm0.51}$ | $38.40_{\pm0.26}$ | $39.57_{\pm1.04}$ |

Table 9: Accuracy evaluated on the bias-conflicting samples for the Colored MNIST and Corrupted CIFAR-10[1,2] datasets with varying difficulty of the bias attributes. We denote bias supervision type by ○ (no supervision), ◐ (bias-tailored supervision), and ● (explicit bias supervision). Best performing results are marked in bold.

| Dataset | Difficulty | Vanilla ○ | Ours ○ | HEX ◐ | REPAIR ● | Group DRO ● |
|---|---|---|---|---|---|---|
| Colored MNIST | 1 | $51.32_{\pm0.45}$ | $68.03_{\pm1.11}$ | $50.54_{\pm0.88}$ | $67.70_{\pm1.02}$ | $\mathbf{68.77}_{\pm1.26}$ |
| | 2 | $45.54_{\pm0.65}$ | $\mathbf{75.56}_{\pm1.22}$ | $46.84_{\pm0.44}$ | $63.71_{\pm0.29}$ | $69.28_{\pm1.13}$ |
| | 3 | $45.48_{\pm1.29}$ | $\mathbf{74.29}_{\pm1.78}$ | $47.88_{\pm1.37}$ | $70.05_{\pm2.10}$ | $68.68_{\pm1.26}$ |
| | 4 | $44.83_{\pm0.18}$ | $\mathbf{74.19}_{\pm1.94}$ | $46.96_{\pm1.20}$ | $68.26_{\pm1.52}$ | $69.58_{\pm1.66}$ |
| Corrupted CIFAR-10[1] | 1 | $44.23_{\pm2.61}$ | $43.76_{\pm1.16}$ | $35.50_{\pm2.82}$ | $38.11_{\pm0.68}$ | $\mathbf{57.34}_{\pm1.33}$ |
| | 2 | $28.47_{\pm0.63}$ | $\mathbf{49.04}_{\pm2.08}$ | $16.82_{\pm1.01}$ | $36.81_{\pm0.93}$ | $45.03_{\pm1.66}$ |
| | 3 | $21.71_{\pm3.37}$ | $\mathbf{44.22}_{\pm2.69}$ | $13.46_{\pm0.41}$ | $35.15_{\pm1.88}$ | $39.64_{\pm1.91}$ |
| | 4 | $14.24_{\pm1.03}$ | $\mathbf{39.55}_{\pm2.56}$ | $8.37_{\pm0.56}$ | $33.05_{\pm0.36}$ | $28.04_{\pm1.18}$ |
| Corrupted CIFAR-10[2] | 1 | $30.56_{\pm0.39}$ | $\mathbf{46.27}_{\pm2.66}$ | $22.51_{\pm0.45}$ | $41.19_{\pm0.32}$ | $43.64_{\pm1.45}$ |
| | 2 | $24.70_{\pm1.03}$ | $\mathbf{44.74}_{\pm3.01}$ | $19.37_{\pm0.41}$ | $39.99_{\pm0.57}$ | $40.94_{\pm0.34}$ |
| | 3 | $19.77_{\pm1.44}$ | $\mathbf{41.72}_{\pm1.90}$ | $16.14_{\pm0.28}$ | $38.41_{\pm0.48}$ | $40.61_{\pm2.01}$ |
| | 4 | $12.11_{\pm0.29}$ | $\mathbf{40.84}_{\pm2.06}$ | $5.11_{\pm0.59}$ | $38.81_{\pm0.20}$ | $37.07_{\pm1.02}$ |

**Difficulty of the bias attribute.** In addition to varying ratio of bias-conflicting samples, we vary the level of "difficulty" for the biased attributes by controlling how much the target attribute is easy to distinguish from the given image. Based on our observations in Section 2, the difficulty of bias is lower, the more likely the classifier is to suffer from bias. We introduce four levels of difficulty for the Colored MNIST and Corrupted CIFAR-10[1,2] datasets with the target and the biased attribute chosen as (Digit, Color) and (Object, Corruption), respectively. For the Colored MNIST dataset, we vary the standard deviation of the Gaussian noise for perturbing the RGB values of injected color. In cases of Corrupted CIFAR-10, we control the "severity" of Corruption, which was predefined by Hendrycks and Dietterich [12]. We provide a detailed description of difficulty of the bias attribute in Appendix A.

In Table 8, 9, we again observe our algorithm to consistently outperform the baseline algorithm by a large margin, regardless of the difficulty for the biased attribute. Furthermore, we observe that both baseline and LfF trained classifiers get more biased as the difficulty of the bias attribute increase in general, which also validates our claims made in Section 2. Notably, LfF even outperforms the baseline methods that utilizes explicit label on the bias attribute in most cases.

Table 10: Accuray evaluated on the unbiased samples and bias-conflicting samples for the Colored MNIST datasets with varying ratio of bias-aligned samples.

| Dataset | Ratio (%) | Unbiased | | | Bias-conflicting | | |
|---------|-----------|----------|------|------|------------------|------|------|
| | | Vanilla | Ours | RUBi | Vanilla | Ours | RUBi |
| Colored MNIST | 95.0 | $77.63_{\pm 0.44}$ | $\mathbf{85.39}_{\pm 0.94}$ | $78.22_{\pm 0.34}$ | $75.17_{\pm 0.51}$ | $\mathbf{85.77}_{\pm 0.66}$ | $75.84_{\pm 0.36}$ |
| | 98.0 | $62.29_{\pm 1.47}$ | $\mathbf{80.48}_{\pm 0.45}$ | $64.92_{\pm 0.78}$ | $58.13_{\pm 1.63}$ | $\mathbf{80.67}_{\pm 0.56}$ | $61.04_{\pm 0.83}$ |
| | 99.0 | $50.34_{\pm 0.16}$ | $\mathbf{74.01}_{\pm 2.21}$ | $52.41_{\pm 0.42}$ | $44.83_{\pm 0.18}$ | $\mathbf{74.19}_{\pm 1.94}$ | $46.85_{\pm 0.46}$ |
| | 99.5 | $35.34_{\pm 0.13}$ | $\mathbf{63.39}_{\pm 1.97}$ | $36.42_{\pm 0.37}$ | $28.15_{\pm 1.44}$ | $\mathbf{63.49}_{\pm 1.94}$ | $29.36_{\pm 0.43}$ |

**Comparison to other combination rule.** There have been several works that utilize intentionally biased models to debias another model. RUBi proposed by Cadene et al. [3] masks original prediction with the mask obtained from the prediction of the biased model. LearnedMixin proposed by Clark et al. [5] uses an ensemble of logits of two models. DRiFt proposed by He et al. [10] learns residual of the pretrained biased model to obtain the debiased model. As an effort to keep the usage of human knowledge minimal, we designed our combination rule without any hyperparameter. In Table 10, we constructed an ablation study on LfF with a combination rule replaced by that of RUBi. While our method equipped with the RUBi combination rule slightly improves accuracy over the vanilla model, it is far behind other resampling/reweighting based methods like REPAIR and Group DRO. In conclusion, resampling/reweighting based methods are generally effective method than manipulating the predictions or logits directly.