[Reviews · NeurIPS 2020]

Review 1

Summary and Contributions: This paper addresses the issue of bias in classifiers, i.e. reliance on spurious correlations. The authors posit that such bias takes a malignant form only if learning a spurious correlation is *easier* than learning a desired property. They state that the networks focus on easy samples in the early stages of training, making the loss trajectories of the bias-aligned samples distinct from the loss trajectories of the samples that contradict the bias. This motivates their approach of training two models: the *biased* model which aims to amplify the bias using the generalized cross entropy loss, and the *debiased* model, which relies on the biased model to focus on harder samples (that contradict the bias). The authors carry out controlled experiments on Colored MNIST and Corrupted CIFAR as well as on real world datasets, CelebA and BAR (biased action recognition, proposed in this paper). They outperform several state-of-the-art models in the controlled experiments and perform on par on CelebA. Moreover, the proposed method does not require expensive supervision on the bias, whereas the state-of-the-art does. UPDATE I appreciate the clarifications and additional results provided by the authors, and still think that the paper should be accepted. However, in light of several issues raised by the other reviewers, in particular, about the claims re available supervision, I have decreased my score to 7.

Strengths: The paper is overall clearly written. The explored direction is definitely of interest to the NeurIPS community. The authors present a clear intuition for their approach to classifier debiasing, supported by the observed learning curves and experimental results. The authors abate the generalized cross entropy loss, showing its effectiveness. The proposed method does not require expensive supervision on the bias. The experiments are conducted on two synthetic datasets as two real-world ones. Moreover, the authors contribute a new realistic dataset, BAR (biased action recognition).

Weaknesses: All experiments seem to involve a pair of correlated attributes (one biased w.r.t. another), but how will this method handle more than 2 attributes with multiple biases at once? Figure 4: the authors highlight that the biased/vanilla models underperform on the bias-conflicting sets, but do not discuss that the *debiased* model is doing poorly on the bias-aligned sets, especially for Corrupted CIFAR-10 (only about 60% accuracy). Does this reflect a trade-off relationship, i.e. one has to lose accuracy on the bias-aligned samples in order to do well on the bias-conflicting samples? Do the authors have some thoughts on this? Would have been great to see more comparison to the state-of-the-art methods on CelebA and, especially, on the BAR dataset.

Correctness: The claims and method appear correct, to the best of my understanding.

Clarity: L85: what is H() here? L203-217: this discussion presented here is specific to the Colored MNIST, not the Corrupted CIFAR-10, where the numbers are much lower. Table 3: the bias-conflicting results are not discussed.

Relation to Prior Work: The authors should clarify what specific bias-related supervision the prior works have access to. That would help better inform the reader early on.

Reproducibility: Yes

Additional Feedback: The authors mention the connection of their work to Aprit et al. [1], who point out that the neural networks learn easier concepts earlier than the hard ones. With that in mind, it seems that the proposed scheme should be applicable to improve learning on imbalanced datasets / datasets with long-tailed distribution in general. Have the authors tried any such experiments? The authors frequently use the word “settle”, which often seems out of place and better be replaced with a different word. L165,174: compare => compared L174: result => resulting L180: outperforms => outperform Table 3 caption could mention that these results are with 99% bias-aligned training data points. L255: Table 6 => Table 5


Review 2

Summary and Contributions: The paper addresses the dataset bias problem: models, when trained naively, may learn to only capture easy shortcuts for the problem, which only rely on spurious cues rather than the actual (causal, if you like) connections to the target task. The method, Learning from Failure (LfF), relies on the assumption that certain bias cues are learned more quickly than the others. LfF trains two networks f_B and f_D. f_B is designed to amplify its own prediction, thereby amplifying its reliance on bias cues (according to our assumption). f_D then learns a different representation from f_B by focusing more on samples that f_B fails to recognize, assuming they are less bias-inflicted. The paper contributes: - A new methodology for reducing bias in models (albeit the existence of similar frameworks that are not discussed). - An empirical observation that some biases are more quickly captured during training (albeit with questionable generalizability).

Strengths: 1. The paper tackles an important issue. The paper also aligns well with the re-emphasized objective in this NeurIPS round - the societal impact of the paper. The task itself is directly applicable to e.g. the correction of demographic biases in ML models. 2. The paper presents a good attempt at analysing the different learning curves for different types of cues. For example, color is more quickly learned than digit for Colour MNIST. Expanding this analysis would be an interesting way to improve the impact and significance of the paper. 3. The method is simple and easy to apply on a diverse set of domains (vision, language, speech, etc.), albeit being limited to the classification task (due to the use of generalised cross entropy as the loss function).

Weaknesses: 1. LfF does use human knowledge; please do not claim that it doesn't. The paper criticises prior works on the de-biasing problem for using "domain-specific knowledge" or "explicit supervision" on the suprious correlated attributes, while claiming their methods to be designed for scenarios where "such information is unavailable". I strongly disagree with this bold claim. LfF heavily depends on the assumption that the quickly-learned cues (so-called "malignant biases") are the undesired biases that hinders generalisation. Do quickly-learned cues **always** correspond to undesired set of biases? I don't think so. In Table 1, colour is more quickly learned than digits. However, is colour always the undesired bias and digit the desired causal cue? Well, it depends on the task. If you really want to train a good colour classifier, then the digit cue may be a nuisance. This means - you need to have the knowledge that "colour is an undesired bias in my task" and that "colour is more quickly learned than digits". These are exactly the "domain knowledge" the paper is criticising. The authors may argue that the LfF can still be blindly applied to any recognition task (why not?) and we can expect performance gains, whether the undesired bias is quickly or slowly learned for an average ML model. About this, I fully agree that the framework itself is free of conditioning on the particular bias type and this is a great advantage that it has over certain previous methods. However, the evaluation and the model selection procedure are conditioned on the human-defined bias types in this paper. Colour MNIST uses the desired-undesired cue pair (digit, colour); CIFAR uses (object, corruption); CelebA uses (HairColor/HeavyMakeup, Gender); BAR uses (action, place). One may say they are just evaluations and they do not have to do with the claim that the method is independent of human-defined bias types. Well, one should be careful about this point. Lots of implicit model selection and design choices are made through evaluation over the validation set. Since we use the human-defined bias types in evaluation, the implicit design choices for LfF are inevitably geared towards improving the generalisation performances across those specific types of biases. It is very difficult, if not impossible, to let a model learn in a truly "unsupervised" manner, as long as system is evaluated on a dataset that comes with GT labels. Please describe what efforts have been put in terms of experimental setup to ensure that the method is truly designed to be independent of the human-defined bias type. How are the hyperparameters in the system chosen? Will they transfer well to unseen data, task, and architectures? If so, why? 2. Deeper analysis needed. While I really like the observation that there exist discrepancies in the speeds different cues are learned, I wonder if the observation is generalisable. It would have been nicer if the observation is verified over multiple tasks, data, and architectures. It would also be nice to include more analysis and characterisation of cues that are more quickly learned than the others. E.g. why are colour and corruption cues more quickly learned than digit and object classes? Can you make a general statement about what makes a cue more quickly learnable?

Correctness: I wish there is more discussion about how specific design choices are made in the model, as well as how the val and test splits are formed, to ensure that the design choices do not overfit to the validation set (where you implicitly exploit the human-defined bias information).

Clarity: 1. Malignant/benign bias. The "malignant" and "benign" biases in this paper actually mean "quickly-learned" and "slowly-learned" biases, respectively. I think the choice of words is not ideal because it is not always the case that more quickly-learned biases are undesirable (see weakness #1). 2. Clean up notations in Sec 2.1. I do not understand this section. Are a_1,....,a_k integers? What does y=a_t mean? Are they random variables? What is H? Is it mutual information? Conditional entropy? What do the subscripts mean? 3. What is the value of q in different setups? If I have not mistaken, the actual values of q in experiments are never defined in the main paper or Appendix. Please let us know which values it takes for different setups and how the values are chosen - through validation?

Relation to Prior Work: Another big con against the paper is that it misses a discussion of a whole set of related work on model de-biasing. The LfF combination rule in Equation in L168 (please attach equation numbers) has many similarities to below works, both in terms of motivation and in terms of methodology. 1. RUBi: Reducing Unimodal Biases for Visual Question Answering. NeurIPS 2019. - See Equation 4 and Figure 4d. 2. Don’t Take the Easy Way Out: Ensemble Based Methods for Avoiding Known Dataset Biases. EMNLP-IJCNLP 2019. - See Section 3.2.4. 3. Unlearn Dataset Bias in Natural Language Inference by Fitting the Residual. EMNLP-IJCNLP 2019. - See Equation 5. 4. Look at the First Sentence: Position Bias in Question Answering. arXiv 2020. & Training products of experts by minimizing contrastive divergence. Neural computation. Neural Computing 2002. - See Equation 1. 5. Learning De-biased Representations with Biased Representations. ICML 2020. - See Equation 2. In all the papers above, an intentionally biased model guides the learning of the target model to be de-biased. Please check the specific implementations in the pointers above and answer the below three questions: - How does LfF combination rule compare against those prior works? - Conceptually, why should we expect LfF combination rule to be better than those combination rules? - Empirically, does LfF combination rule work better than the existing combination rules? Authors may argue that the core contribution is the utilization of training speed differences among cues, but the combination rule is another important ingredient for a de-biasing method. Since the paper is proposing a potentially novel combination rule, please perform a valid comparison against prior works.

Reproducibility: Yes

Additional Feedback: ***POST REBUTTAL*** I generally think the authors have address the concerns well. Please see below. > LfF depending on human knowledge This was the big con against this paper I had because the paper was misleading the readers to consider LfF (proposed method) as free of human knowledge, while other methods are not. Such a description is unfair to the stream of de-biasing papers which have explored different sources of including bias supervision than explicit bias labels. LfF, in my view, is using yet another source of human knowledge on bias. The authors have addressed this concern well by saying "we will further clarify the type of knowledge used by LfF and prior work". If this is (going to be) explaned fairly in the final version, I don't have any major con against the paper anymore. > Generalization of the observation. I appreciate the extra experiments the authors have performed, but what I was asking for was a more precise characterisation of slowly-learned and quickly-learned features by CNNs, rather than the hand-wavy descriptions like "local patterns" and "high-level concepts". But this is not a major point. > Relation to prior work with combination rules. I'm glad that the authors have compared against the RUBi combination rule. It would have been better if the authors have also compared against LearnedMixin, for example, which has a much simpler combination rule and yet is performant. But again, I wouldn't reject a paper because of this.


Review 3

Summary and Contributions: This paper propose a new learning algorithm to combat the performance degeneration raised from training on the biased training set. The key idea is to use the training loss of bias amplified model on the data instance to compute the loss weight for an unbiased model. To obtain such a bias amplified model, this paper uses the phenomenon that bias-aligned examples are easier to learn, and therefore use the generalized cross-entropy objective (which amplifies the training loss of examples that are highly confident) to amplify the bias in learning of such a biased model. Next, with this learned biased model, this paper further learns an unbiased model by using the biased model to measure how an example is aligned with the dataset's bias. Intuitively, It down weighs the data samples that are more biased and up weighs the sample that are less biased.

Strengths: The overall design of the learning algorithm is interesting and well-motivated. The arguments made are very well supported with empirical results and they step-by-step builds the designed algorithm. Very well written. The authors first verified the phenomenon in learning from biased data, where bias-aligned examples (easy) got first learned by the model comparing to bias-conflicting examples. Then it designs the bias amplified learning using the generalized CE objective to obtain a biased learner, which proves to be very important to the final success of the overall method. The algorithm learns to combat bias without making explicit assumption about what the bias is, which makes it very generally applicable. Empirical performances on both synthetic dataset and real-world dataset are strong.

Weaknesses: I am curious what would be the outcome when applying the proposed algorithm to unbiased training set. The biased neural network in this case will try to pickup the easy examples and therefore the it does some sort of hard-negative sampling for the un-biased neural network? This could be interesting to explore. In the experiments, there are a few points I have questions: 1. In table 4, it seems that Group DRO is better at learning the bias-conflicting examples than the proposed method? You have explained that possibly due to the fact that Group DRO is explicitly maximizing the worst-case group accuracy. But I think you are doing similar things? Basically performing weighing the bias-conflicting examples more than bias-aligned examples. 2. BAR evaluation set results. It would be impossible to understand how well LfF without having some reasonable baseline algorithms on the proposed BAR benchmark. I would strongly recommend to also run Group DRO and other algorithms on this setting.

Correctness: Yes.

Clarity: Yes.

Relation to Prior Work: Yes.

Reproducibility: Yes

Additional Feedback: Post rebuttal feedback. After reading the author response and other reviewers' comments, I believe that most of my concerns are addressed (i.e. requesting additional baseline results on BAR benchmark). Therefore, I would like to rise my score.


Review 4

Summary and Contributions: This paper studies an important problem in machine learning: learning can be biased by unintentionally correlated attributes. This paper also makes an interesting observation that biased attributes only get remembered when they are “easier” to learn than normal data attributes. This paper proposes a simple but effective debiasing training scheme to train debiased DNNs in the presence of biased data attributes. The proposed training scheme does not need prior knowledge on the bias, and is verified on both synthetic and real-world datasets. ----------- I raised my rating to 6 after the rebuttal.

Strengths: - Tackling an important but challenging problem with a novel training strategy. - A comprehensive analysis of the bias problem on existing datasets, and an interesting observation: biased data can only be learned when they are ‘’easier’’. - The proposed training strategy seems generic and effective. Obvious improvements over existing debiasing methods. - Well-controlled experiments and detailed analysis.

Weaknesses: - The grouping of bias into malignant vs benign seems not very convincing. This is because training with those biases can vary under different settings, for example, the number of training samples, the number of training epochs, learning rates, weight decay and model architectures. The biased attribute can be malignant under one setting, while is benign under a different setting. What would happen to a benign bias attribute when the total number of input dimensions are reduced? For example, just leave the center area of the Color MNIST images? - I cannot agree with the claim “easier” bias attributes are likely malignant. To me, it is more like “the texture biases (like color) are likely malignant, while the shape biases are likely benign”. It is hard to say “easier to learn”, but more like “learn more thoroughly”, as the color information will contribute the most to the neurons’ activations in the early stage. - The use of GCE loss is not well motivated, as there are many loss functions that differentiate between low and high confidence samples including the CE loss. GCE was initially designed to mitigate overfitting to noisy labels, however, it seems that the baised model needs more overfitting (to the bias attribute)? Suppose the upweight to CE gradients is indeed necessary, then why not use Focal Loss and its variant? Another question is, since the biased model learns the bias in the early stage, then why not stop training it at the later stages, considering that the later stage will learn more hard (not easy) examples, as pointed out in Arpit et al. [1]. Isn’t the later training will forget some easy examples, or equivalently ‘easier’ and malignant attributes? - Why need two networks? If you look at the two equations at lines 162 and 168, the confidence output of the network can also be used to do the weighting? In other words, since ‘easier’ (e.g. high confidence) examples can carry malignant bias, so not just use the confidence to obtain the relative difficulty score? Or why not modify the GCE loss to avoid training a biased model? Why it has to train an extremely biased model using GCE, then use it to train a second model. The proposed two-network architecture looks similar to co-teaching [2] and related structures, but not discussed.

Correctness: Debatable.

Clarity: Can be improved.

Relation to Prior Work: Some missing relation to prior work.

Reproducibility: Yes

Additional Feedback:

[Author Response · NeurIPS 2020]

We sincerely thank all reviewers for the insightful comments and feedback on our work of learning from failure (LfF).

**[R1-1] Multiple bias attributes.** One of the strengths of LfF is that it does not require an explicit characterization of
bias, enabling the handling of multiple/composite biases by default. Indeed, the CelebA dataset contains a diverse set of
attributes that may be spuriously correlated, but LfF performs consistently well, as Table 4 suggests.

**[R1-2] Trade-off of debiasing.** We *do not* interpret this as a "true" trade-off, as debiasing does not degrade the model's
ability to capture the desired correlation; indeed, the performance of LfF is similar on bias-aligned and bias-conflicting
sets. Instead, we view the apparent underperformance as a result of "not utilizing a (delusional) spurious correlation."

**[R1-3] Comparison to SOTA on BAR.** Following R1's suggestion, we additionally test ReBias [2] (SOTA among
bias-label-free methods) on BAR, using the official code (released after the submission deadline). ReBias achieved
59.73% accuracy, which was lower than 62.98% achieved by LfF; this result will be added to Table 5.

**[R2-1] LfF depending on human knowledge.** We do not claim LfF to be free of human knowledge and will further
clarify this in the final draft. As R2 pointed out, LfF leverages a yes/no type of knowledge: on the given setting, the
bias is learned faster than the desired correlation. This is also consistent with our claim that LfF is not "domain-specific"
since we followed prevalent use of the word "domain" [2, 22, 29], i.e., groups with the same data types (e.g., natural
image) or bias types (e.g., texture). However, this consistency may not hold depending on the definition of "domain."

Hence, we deeply resonate with R2's concern, and we will further clarify the type of knowledge used by LfF and
prior work. We will describe (a) LfF as utilizing the aforementioned yes/no type of knowledge and (b) the previous
works as utilizing explicit labels or conditioning on the particular bias type, instead of using the term "domain-specific"
knowledge. For example, we will modify L2-5 in the abstract by "In this work, we propose a new algorithm utilizing a
new yes/no type of knowledge, which does not use explicit labels or presume a particular bias type."

For *empirical evaluation*, we inevitably use additional human knowledge for choosing the attribute pairs, as R2
mentioned. However, we only use the LfF's yes/no type of knowledge for choosing one of the attributes as an undesired
bias, e.g., we set `Color` as an undesired bias since it is "easier" than `Digit` for a vanilla model. For *model selection*,
we put significant effort into making LfF rely less on human knowledge. Existing hyperparameters, except for GCE
parameter $q$, are obtained from training a vanilla model (using popular architectures) without any prior consideration of
the bias. The GCE hyperparameter $q = 0.7$ is simply taken from the original paper [25].

**[R2-2] Generalization of the observation.** Following R2's suggestion, we further verify
our observations' generalizability with the 3D-shapes dataset. We repeat the experiments
in Figure 2 using (`ObjectScale`, `WallHue`) attributes (see right). This observation aligns
with [27]; CNNs are good at learning local patterns rather than the high-level concepts.

**[R2-3] Relation to prior work with combination rules.** As an effort to keep the usage of human knowledge minimal,
we designed our combination rule without any hyperparameter (in contrast to [2, 28, 29]) though it is not our main
contribution. As R2 suggested, we constructed an ablation study on LfF with a combination rule replaced by that of
RUBi. Our LfF combination rule achieves 74.01% while that of RUBi achieves 52.41% on the Colored MNIST dataset.
We will add more discussions and experiments in the final draft.

**[R4-1] Comparison with Group DRO & BAR baseline.** While Group DRO assumes and exploits the availability of
*worst-case group labels*, LfF achieves a similar or better performance without requiring such additional labels (Tables
2 and 4). This difference makes LfF applicable on datasets without group labels (such as BAR), while Group DRO
cannot. Instead of Group DRO, we provide an additional BAR baseline using ReBias [2]; see **[R1-3]**.

**[R7-1] Malignancy of bias depending on the algorithm.** Our definition of "malignant bias" (see L119) is consistent
under the scenarios R7 suggested; we define a spurious correlation as malignant whenever the existence of the correlation
leads to a performance degradation *under the given setup*, not as a "global" characteristic.

**[R7-2] Alternative word for "easier".** We describe the bias as "easier" when it is learned during the early stage,
following [1, 31]; still, the expression "learned more thoroughly" which R7 suggested would be appropriate as well.

**[R7-3] Alternative schemes for focusing on easy samples.** Focal loss is devised to focus on "hard" examples, and
thus it cannot replace the GCE loss, which focuses on "easy" examples. Early stopping is not needed for the biased
model trained with GCE loss since it remembers the easy examples, even in the later stages (see Figure 4).

**[R7-4] Alternative schemes for weighting samples with two networks.** We follow the common practice [30] using
two networks to distinguish sample groups by leveraging the implicit bias of neural networks. Such practice is known
for its high-performance (often better than using one network). We will add the suggested reference in our final draft.

[27] W. Brendel and M. Bethge. Approximating CNNs with bag-of-local-features models ... in ImageNet. ICLR, 2019.
[28] C. Clark, et al. Don't take the easy way out: ... known dataset biases. EMNLP-IJCNLP, 2019.
[29] H. He, et al. Unlearn dataset bias in natural language inference by fitting the residual. EMNLP-IJCNLP, 2019.
[30] J. Li et al. Dividemix: Learning with noisy labels as semi-supervised learning. ICLR, 2020.
[31] S. Sagawa et al. An investigation of why overparameterization exacerbates spurious correlations. ICML, 2020.


[Meta-Review · NeurIPS 2020]

The paper makes an interesting observation that cometimes, the classifier "biases" can be detected from examples that the models corectly classify early on. They present the first successful unsupervised method to correct for such biases, with promising experimental results since it compares reasonably compared to supervised approaches on artificial and real datasets. Moreover, the reviewers noted that the paper tackles a very timely and important topic.